# Development and Validation of a simple score for diagnosis of Leptospirosis at outpatient departments

**Nidhikul Temeiam[1]☉, Sutthi Jareinpituk[1]☉, Phichayut Phinyo[2]‡, Jayanton Patumanond[2]‡, Nattachai Srisawat**[ORCID][3,4,5,6]*

**1** Department of Epidemiology, Faculty of Public Health, Mahidol University, Thailand, **2** Center for Clinical Epidemiology and Clinical Statistics, Faculty of Medicine, Chiang Mai University, Thailand, **3** Division of Nephrology, Department of Medicine, Faculty of Medicine, Chulalongkorn University, and King Chulalongkorn Memorial Hospital, Bangkok, Thailand, **4** Tropical Medicine Cluster, Chulalongkorn University, Bangkok, Thailand, **5** Critical Care Nephrology Research Unit, Faculty of Medicine, Chulalongkorn University, Bangkok, Thailand, **6** Academic of Science, Royal Society of Thailand, Bangkok, Thailand

☉ These authors contributed equally to this work.
‡ PP and JP also contributed equally to this work.
* drnattachai@yahoo.com

**Data Availability Statement:** All relevant data are within the manuscript and its Supporting Information files.

## Abstract

### Background

Leptospirosis is an important zoonotic disease within the tropics. Diagnosing leptospirosis is a clinical obstacle, as clinical presentations are similar to other tropical infectious diseases. Available serological tests are often insensitive and not cost-effective. Many clinical diagnostic scorings had been developed but most were based on hospitalized patients, and wound not be suitable for use in suspected patients in setting of ambulatory care.

### Objectives

To develop and internal validate multivariable diagnostic prediction score of leptospirosis in patients suspicious of leptospirosis at out-patient clinics of community hospitals.

### Materials and methods

We performed a prospective, multisite diagnostic prediction research with development of a diagnostic score. The development cohort was based on patients suspicious of leptospirosis who visited five community hospitals in Si Sa Ket province, Thailand during December 2017 to November 2018. Leptospirosis confirmed cases were defined when one of the three standard confirmatory tests was positive. Multivariable logistic regression was used for score derivation. Test of AuROC equality was done to compare diagnostic performance of the newly derived score and conventional WHO score.

### Results

A total of 262 leptospirosis suspicious patients were enrolled. Eighty-two patients (31.5%) were leptospirosis confirmed cases. Five final predictors remained within the reduced

**Funding:** Nattachai Srisawat received funding from Jongkolneenithi Foundation and Medical Association of Thailand. The funders had no role in study design, data collection and analysis, decision to publish, or preparation of the manuscript.

**Competing interests:** The authors have declared that no competing interests exist.

logistic model which were history of exposure to wet ground at workplace, history of contact water reservoir used by animal, urine protein and urine blood positive from dipstick test, and neutrophil count from CBC $\geq$80%. The OPD score diagnostic performance was AuROC 0.72 (95%CI 0.65–0.79). Test of equality revealed significant differences of AuROC between the OPD and WHO score (0.72 vs 0.62, p-value 0.014). Patients were categorized into low and high probability of having leptospirosis at score point of 3.5 with sensitivity 72.4% and specificity 61.7%.

## Conclusions

This study developed and internal validated the OPD score. This clinical risk score could be one of the important tools for diagnosis of leptospirosis at the outpatient clinic.

### Author summary

Leptospirosis is an important tropical infectious disease. Early diagnosis of leptospirosis in patients with mild and vague clinical syndrome is another clinical obstacle. Most of the diagnostic score developed for diagnosis of leptospirosis are based on patients with flank clinical symptoms, mostly in hospitalized patient. This study developed the OPD score for early diagnosis of undifferentiated fever for patients visiting outpatient care in leptospirosis endemic area. This score can be practically apply to area where health care facilities are limited by asking patients for only two potential risk factors of exposure to *Leptospira*, taking simple blood and urine samples. We believed that this score could aid physician in early diagnosis and initiation of treatment in early leptospirosis patient which would alleviate disease progression and probably decrease mortality.

## Introduction

Leptospirosis is an emerging tropical zoonotic disease caused by a genus of spiral-shaped bacteria, *Leptospira*. Humans usually acquired the disease from environmental exposure to the organism shed from the urine of mammal hosts.[1–3]. In Thailand, the prevalence of leptospirosis was estimated at 5 to 9 cases per 100,000 population with mortality rate at 5 to 10 percent [4]. Si Sa Ket province, a northeastern province with 70 percent of population at risk from occupational exposure [5], reported the highest incidence of 18.7 to 28.2 per 100,000 populations from 2011 to 2015 and also carried three times higher case fatality rate than national averages [6–8].

Diagnosing leptospirosis is a clinical challenge, as the initial presentations are usually difficult to distinguish from other tropical infectious diseases [9–11]. Spectrum of the disease varies from subclinical case, mild case to severe case which could be fatal. Therefore, early detection and early treatment in suspicious groups of patients are vital to attenuate disease progression [1, 11–13]. At present, several options of laboratory investigations are available but each carries its own limitation, especially in countries with limited resources. Bacterial isolations from specimen culture are less sensitive and the results are delayed [11, 14]. Serological testing such as microagglutination test (MAT) needs two sets of blood samples for interpretation of the result and only few laboratories are accessible [15–17]. Rapid antibody test is proved to be non-sensitive and non-specific especially in the early phase of the infection [14, 18–20].

Polymerase chain reaction (PCR) requires specialized laboratories and technicians which is not applicable and not cost-effective [1, 11, 14].

Primary care physician relies primarily on clinical characteristics, signs and symptoms of the patients for diagnosing leptospirosis. Faine's criteria or WHO score and modified version in 2004 and 2012, were developed as diagnostic guide for clinicians to make presumptive diagnosis of leptospirosis [21–24]. As they were developed from hospitalized patients cohort, the criteria seems to be appropriate for patients with higher degree of severity which require hospitalizationand hospitals where serological investigation are available. However, whether the use of such criteria is suitable for implementation in primary care setting, where patients usually presented in early phase with vague clinical syndrome, is still questionable.

This study aimed to develop a practical diagnostic tool for leptospirosis that incorporates clinical signs and symptoms, history of exposure to possible risks, and routinely available laboratory data to aid physicians and associated health care workers in community hospitals within an endemic area for early detection and early treatment of the disease.

## Methods

### Ethics statement

The study protocol was approved by the ethical review committee for human research, of The Faculty of Public Health, Mahidol University (MUPH 2017–204), and the ethical committee for research in human subject of Si Sa Ket Hospital (COA No.004 REC No. 071/2560). All study participants were requested for informed consents prior to study inclusion.

### Study design and setting

A diagnostic prediction research and clinical diagnostic score development was performed. The data was prospectively collected from five community hospitals with top highest prevalence of leptospirosis within Si Sa Ket Province (Khukhan, Khun Han, Phu Sing, Phrai Bueng, and Prang Ku Hospitals).

### Study participants

Patients suspicious of leptospirosis by physicians from initial clinical presentation who visited out-patient clinic in each study site during December 1, 2017 to November 30, 2018 were asked for consents and subsequently included into study. Clinicians suspected the diagnosis of Leptospirosis based on typical syndrome stated in the classic WHO clinical criteria such as the presence of acute febrile illness (onset of fever less than 14 days), headache, myalgia with history of exposure to animal water reservoirs or flooded environments either at home or at work [11]. Patients with unstable vital signs requiring resuscitation at first visit, and patients who were unable to communicate with Thai or local language were excluded and were not interviewed. Patients whose disease progressed and did not survive for the second blood sample collection were excluded from analysis.

### Data collection

Enrolled patients were interviewed with standardized questionnaire by trained research personnel. Data collected consists of 5 components which are (1) baseline characteristics and demographic: gender, age, occupation, comorbidities, drinking habit, duration of living within the area, (2) clinical presentations: chief complaint (e.g. fever, myalgia, headache, calf pain), symptoms (e.g. cough, rhinorrhea, red eye, nausea, vomiting, diarrhea, dyspnea, oliguria, jaundice and hemoptysis), onset and duration of chief complaint, prior treatment or visit to other

health care providers, prior antibiotics prescribed, (3) physical examination: initial vital signs (body temperature, pulse rate, blood pressure, respiratory rate), calf tenderness, conjunctival suffusion, jaundice, and presence of wounds in hand, foot or leg, (4) exposure to possible environmental risk factors within the past month: flooded house compound, contact with animal, contaminated water and soil around workplace, presence of wound on dependent parts, contact with animal water reservoir, and features of work. (5) hematologic and biochemical laboratory findings: complete blood count, urine dipstick test, and renal function test.

## Blood sample and specimen collection

The first set of blood samples were sent for blood culture, real time polymerase chain reaction (PCR) and microscopic agglutination test (MAT). Patients were also scheduled for another visit for the second blood samples collection, which would be used as a paired serum to evaluate rising of MAT.

On the first day of enrollment, 12 ml of blood and 30 ml of urine sample were collected. Blood samples for laboratory investigations other than culture were collected in EDTA tubes and clotted blood tubes. Both plasma and urine sample were centrifuged at 3000g for 10 minutes and were subsequently frozen at -20 degree Celsius until being transferred to the central laboratory. Both samples were stored at -80 degree Celsius until they were taken for analysis. For direct culture of *Leptospira*, a drop of blood and urine was separately inoculated into 4 mL of liquid Ellinghausen-McCullough-Johnson-Harris (EMJH) at 29 degree Celsius for 14 days. Detection for *Leptospira* was done with direct observation via dark field microscopy [26]. We confirmed all isolation of *leptospira* by performing 16S rRNA sequencing. For real time polymerase chain reaction (PCR), 200 μL of whole blood was sampled from EDTA tube. A high Pure PCR template preparation kit (Roche Diagnostics, Germany) was used for DNA extraction with 50 μL elution buffer. The amplification primers for LipL32 gene were LipL32-45F (5'-AAG CAT TAC CGC TTG TGG TG-3') and LipL32-286R (5'-GAA CTC CCA TTT CAG CGA TT-3'). The fluorescent probe was LipL32-189P (FAM-5'-AA AGC CAG GAC AAG CGC CG-3'-TAMRA). The PCR reactions of the samples were performed in a final volume of 20 μL which contained 5 μL of genomic DNA and 15 μL of reaction mix (10 μL of 2X TaqMan Universal PCR Master Mix (Applied Biosystems, Foster City, CA), 1 microliter of each 10 μM primer, 0.4 μL of 10 μM probe under 2.6 μL distilled water). The real time PCR program consisted of 45 cycles, each consisting of 95 degree Celsius for 15 seconds and 60 degree Celsius for one minute. Positive and negative controls were included in every experiment done. Results were read by threshold cycle (Ct) value [27]. Microscopic agglutination test or MAT was performed as described in the standard protocol of the World Health Organization (WHO) guideline [11]. A positive MAT was defined as a single serum cut-point of ≥1:800 based on confirmed laboratory diagnosis by CDC definition 2013 [28]. For all urine dipstick test, the reported results of trace or more (1+, 2+, 3+, and 4+) were considered positive.

## Confirmation of cases [28]

Clinically suspected patients were defined as "Leptospirosis confirmed cases" if one of the following laboratory criteria were met: (1) isolation of *Leptospira* from clinical specimen with confirmation by performing 16S rRNA sequencing (2) *Leptospira* agglutination titer of ≥800by microscopic agglutination test (MAT) in one or more specimens, or four-fold rising of *Leptospira* agglutination titer between acute and convalescent phase (3) detection of pathogenic *Leptospira* DNA by polymerase chain reaction from a clinical specimen. Patients who did not fulfil any of the criteria were classified as "non-cases". The confirmation of diagnosis other than Leptospirosis, in non-cases patients was not done. We defined patients as severe

leptospirosis cases if they required any dialysis support, or required mechanical ventilation support or manifested with clinical jaundice. All laboratory confirmation results were blinded to study site physicians, investigators and research assistances.

## Statistical analysis and study size estimation

Continuous variables were checked for normality and presented with mean and standard deviation for normally distributed data. Median and interquartile range was used for non-normally-distributed data. The differences of means between the two contrast groups were compared using independent t-test or rank-sum test based on normality test. Categorical variables were presented with frequency and percentage. The comparisons of two independent proportions were done with exact probability test or chi-square as appropriate. Univariable logistic regression analysis was done for each potential predictor to explore for its diagnostic performance. The diagnostic odds ratios (dOR) and area under the receiver operating characteristics curves were presented. A statistical significance was declared if two-sided p-values fall below 0.05. Stata statistical software version 15 was used for all analyses. For development of clinical prediction rules, there is currently no standard approach for estimation of study size. The authors reviewed the unpublished data and patient records comparing the clinical characteristics of leptospirosis confirmed cases and non-cases at Si Sa Ket hospital during 2015. The proportion of patients reported exposure to contaminated water was 0.73 and 0.25 for confirmed cases and non-cases of leptospirosis, respectively. Using the comparison of two proportions approach, 12 confirmed cases and 47 non-cases were needed to achieve 80% statistical power and a two-sided alpha error of 0.05. A 10-events-per-variable rule of thumb was suggested by many literatures including the TRIPOD statements for reporting of clinical prediction rules development [29]. For our study, as we planned to include at least 5 potential predictors within the final model, at least 50 confirmed cases were required for model derivation. At confirmed cases: non-cases ratio of 1:4 [30], this study planned to recruit at least 250 patients (50:200).

## Model development

The model was based on complete-case analysis, no data imputation was done. All clinically relevant parameters were included in multivariable logistic regression model to explore for significant predictors of leptospirosis. Backward elimination was done based on both statistical significance from p-value of each predictor and total predictive performance of the model via area under the receiver operating characteristic curve (AuROC). Non-contributing factors with large p-values and lowest magnitude of effect (odds ratio closest to 1.00) were initially eliminated from the regression model. After each predictor was removed, we checked for model diagnostic performance via the AuROC. The predictor was re-entered into the model, if its removal caused substantial decreases in AuROC. The steps were done consecutively until all of the remaining predictors within the model had a p-value of lower than 0.10 on condition that the AuROC of the reduced model must be well preserved. Measure of discrimination and calibration of the final reduced model was done with the use of AuROC curves and Hosmer-Lemeshow goodness of fit test.

## Score derivation and validation

The score was assigned for each of the predictor within the final model based on its logit coefficients. In score transformation, the lowest coefficient of all predictors was used as a denominator, while others were used as numerators. After division of coefficients, the products were rounded up to non-decimal figures. Score was calculated for each patient within the

development cohort. Measure of discrimination and calibration were similarly done for score based logistic regression model. The score was further categorized into two risk groups (low and high probability of having leptospirosis) at an appropriate cut-point. Sensitivity, specificity, positive likelihood ratio and AuROC of each risk category were be displayed. The diagnostic ability of developed score was compared with that of the standard WHO score (conventional Faine's Criteria) with 10 predictors by AuROC equality test and visualized with comparative AuROC curves. We performed internal validation with non-parametric ROC regression with 1,000 bootstrapped sampling.

## Results

### Participants

During study period, a total of 262 patients suspected with leptospirosis were enrolled. Two patients were excluded due to incomplete data, 260 patients were included in score development and internal validation cohort. Of these patients, 82 (31.5%) were confirmed leptospirosis cases (Fig 1). Forty-three cases (52.4%) were treated as outpatient treatment and appointed for subsequent follow-up visit. Thirty cases (40.2%) were admitted for in-patient hospital care.

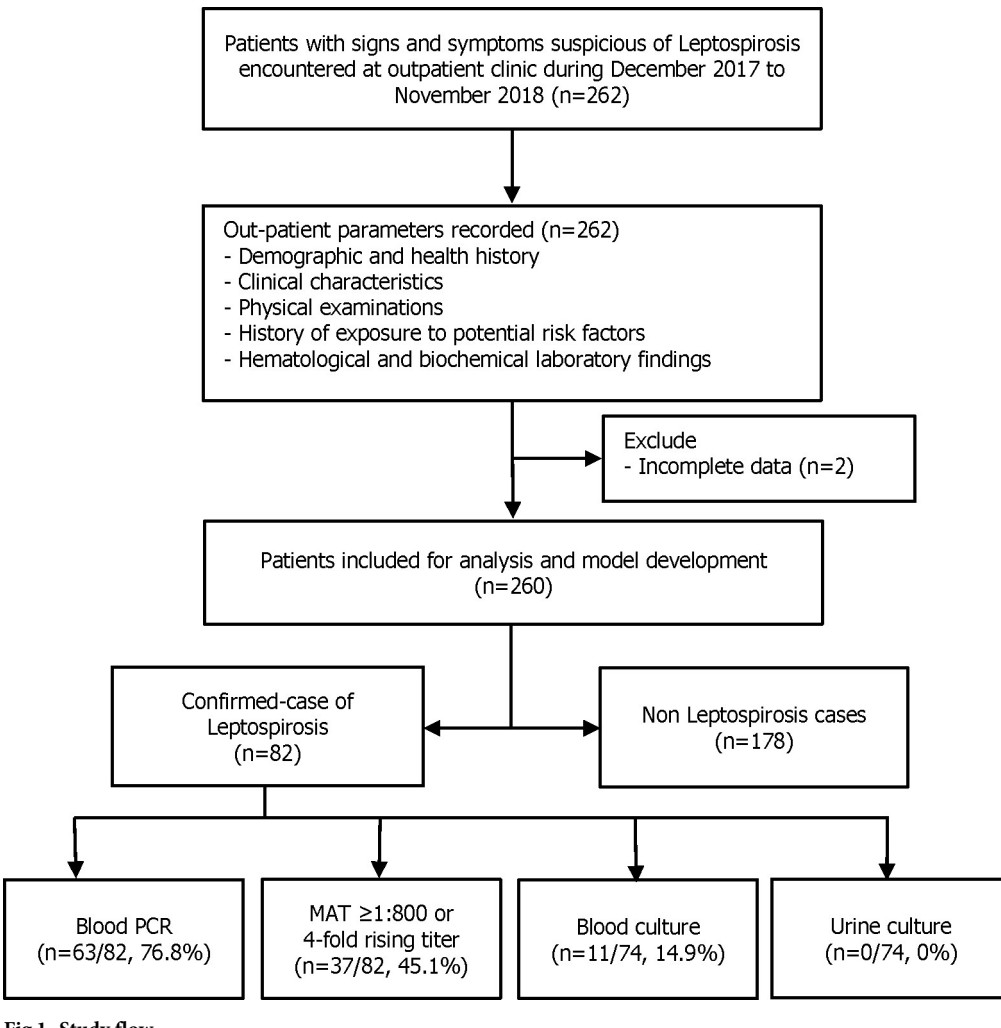

**Fig 1. Study flow.**

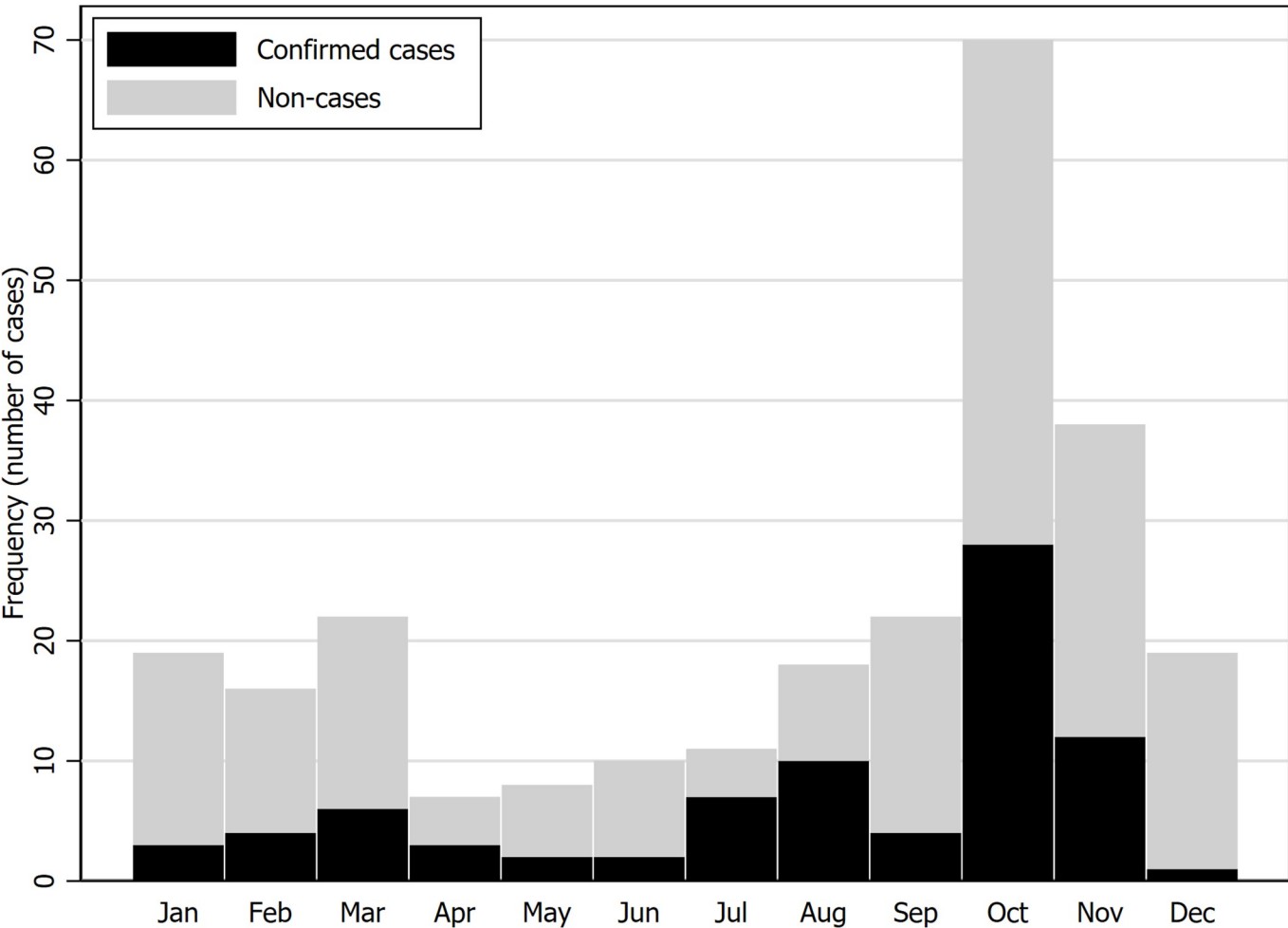

**Fig 2. Epidemiologic curve visualizing distribution and frequency of leptospirosis suspected patients during study period and proportions of cases and non-cases within each month.**

Six cases (7.3%) were defined as severe leptospirosis and were transferred to higher level hospital. However, none of the patient had lethal complication. The monthly epidemiologic curve showed the distribution of suspected leptospirosis cases during study period and the proportion of confirmed cases and non-cases within each month (Fig 2). The number of suspected cases was highest in October and lowest in April. Most of the suspected patients were male (72.3%) without medical comorbidities (84.2%). The mean age of the cohort was 47.0±16.6 years. The mean onset of fever in leptospirosis confirmed cases and non-leptospirosis cases were 3.0±2.3 and 3.2±3.3 days, respectively. Around two third of the patients were farmers. Most patients presented with fever (64.1% vs 72.1%, p = 0.233), myalgia (12.7% vs 18.0%, p = 0.328) and headache (10.3% vs 12.7%, p = 0.675). Most of clinical presentations between both groups were similar. The symptom which showed statistical significance was fatigue (75.6% vs 60.7%, p = 0.024). From physical examination, confirmed cases compared to non-cases had lower systolic blood pressure (115.7±20.2 vs 120.7±15.8 mmHg, p = 0.033), higher pulse rate (100.4±17.6 vs 94.9±16.2, p = 0.011), higher respiratory rate (21.4±2.6 vs 20.7±2.0, p = 0.015), higher proportion of jaundice (6.1% vs 1.1%, p = 0.035) (Table 1).

For history of exposure to environmental risk factors, confirmed leptospirosis cases had higher proportion of flooded house compound (47.5% vs 32.0%, p = 0.025), flooded or wet

**Table 1. Baseline demographic data, presenting symptoms, physical examinations and initial laboratory investigations of the derivation cohort, comparison of confirmed-cases and non-cases of *Leptospirosis*.**

| Clinical Characteristics | Confirmed cases(n = 82) | | Non-cases(n = 178) | | Crude OR (95%CI) | P-value |
|---|---|---|---|---|---|---|
| | n | (%) | n | (%) | | |
| **Demographic** | | | | | | |
| Male | 59 | (72.0) | 129 | (72.5) | 0.97 (0.54–1.75) | 1.000 |
| Age, years (mean±SD) | 45.9 | ±14.6 | 47.6 | ±17.5 | 0.99 (0.98–1.01) | 0.432 |
| **Symptoms** | | | | | | |
| Myalgia | 71 | (86.6) | 141 | (79.2) | 1.69 (0.82–3.52) | 0.172 |
| Jaundice | 7 | (8.5) | 6 | (3.4) | 2.68 (0.87–8.23) | 0.121 |
| Fatigue | 62 | (75.6) | 108 | (60.7) | 2.01 (1.12–3.61) | 0.024 |
| Vomiting | 19 | (23.2) | 27 | (15.2) | 1.69 (0.87–3.25) | 0.120 |
| Breathing difficulty | 14 | (17.1) | 16 | (9.0) | 2.07 (0.96–4.48) | 0.093 |
| **Physical examinations** | | | | | | |
| Body temperature,˚C (mean±SD) | 38.1 | ±1.1 | 37.9 | ±1.1 | 1.20 (0.94–1.54) | 0.141 |
| Pulse rate, per mins (mean±SD) | 100.4 | ±17.6 | 94.6 | ±16.2 | 1.02 (1.00–1.04) | 0.011 |
| SBP, mmHg (mean±SD) | 115.7 | ±20.2 | 120.7 | ±15.8 | 0.98 (0.97–0.99) | 0.033 |
| DBP, mmHg (mean±SD) | 69.5 | ±12.0 | 71.4 | ±10.5 | 0.98(0.96–1.01) | 0.184 |
| Respiratory rate, per mins (mean±SD) | 21.4 | ±2.6 | 20.7 | ±2.0 | 1.15 (1.02–1.28) | 0.015 |
| Jaundice | 5 | (6.1) | 2 | (1.1) | 5.71 (1.08–30.10) | 0.035 |
| **Risk factors** | | | | | | |
| Flood or wet ground at home | 38 | (47.5) | 57 | (32.0) | 1.92 (1.23–3.30) | 0.025 |
| Flood or wet ground at workplace | 71 | (87.7) | 133 | (75.6) | 2.30 (1.09–4.84) | 0.031 |
| Contact animal water reservoir | 34 | (42.0) | 51 | (28.7) | 1.80 (1.04–3.12) | 0.045 |
| **Laboratory findings** | | | | | | |
| WBC, /µL (median, IQR) | 10,000 | 6,600, 12,200 | 7,900 | 5,700, 11,800 | 1.00 (0.99–1.00) | 0.049 |
| Neutrophil count ≥80% | 37 | (45.7) | 41 | (23.3) | 2.77 (1.58–4.85) | <0.001 |
| Neutrophil, % (mean±SD) | 73.3 | ±14.1 | 68.3 | ±13.0 | 1.03 (1.01–1.05) | 0.006 |
| Lymphocyte, %, (median, IQR) | 13.7 | 9.2, 23.6 | 19.9 | 13.1,27.3 | 0.97 (0.94–0.99) | 0.003 |
| Monocyte, % (median, IQR) | 5.1 | 4, 8 | 7.1 | 5.6, 9.1 | 0.86 (0.78–0.94) | <0.001 |
| Platelet, /µL (median, IQR) | 194,000 | 141,000, 271,000 | 214,500 | 181,500, 259,000 | 1.00 (0.99–1.00) | 0.105 |
| eGFR, mL/min/1.73 m$^2$ (mean±SD) | 85.8 | ±23.7 | 90.0 | ±22.9 | 0.99 (0.98–1.00) | 0.187 |
| Urine glucose positive | 8 | (10.4) | 9 | (5.3) | 2.07 (0.77–5.60) | 0.175 |
| Urine protein positive | 44 | (57.1) | 65 | (32.2) | 2.15 (1.25–3.72) | 0.008 |
| Urine blood positive | 38 | (49.4) | 46 | (27.2) | 2.61 (1.49–4.56) | 0.001 |
| Urine bilirubin positive | 7 | (9.2) | 7 | (4.1) | 2.36 (0.80–6.99) | 0.137 |
| Urine ketone positive | 19 | (24.7) | 29 | (17.1) | 1.59 (0.83–3.06) | 0.169 |

Abbreviations: SBP, systolic blood pressure; DBP, diastolic blood pressure; WBC, white blood cell count; eGFR, estimated glomerular filtration rate; OR, odds ratio; CI, confidence interval; SD, standard deviation; IQR, interquartile range.

ground at workplace (87.7% vs 75.6%, p = 0.031) and contact with water reservoir used by animal (42.0% vs 28.7%, p = 0.045). Other factors such as animal contact, presence of wound or skin abrasion and features of work did not reveal significant differences (S1 Table).

Confirmed leptospirosis cases had significantly higher percentage of neutrophil (73.3±14.1 vs 68.3±13.0, p = 0.006), lower percentage of lymphocyte (13.7(IQR 9.2, 23.6) vs 19.9 (IQR 13.1, 27.3), p = 0.003) and monocyte (5.1 (IQR 4,8) vs 7.1 (IQR 5.6,9.1), p<0.001) than non-

cases. The result of urine dipstick test showed significant differences between groups in presence of urine protein (57.1% vs 32.2%, p = 0.008), urine blood (49.4% vs 27.2%, p = 0.001), urine bilirubin (9.2% vs 4.1%, p = 0.137), and urine ketone (24.7% vs 17.1%, p = 0.169). (Table 1). The remaining diagnostic predictors without statistical significant differences from univariable comparison were shown in S1 Table.

## Model development and validation

All potential predictors with p-value from univariable analysis less than 0.2 were included in multivariable logistic regression for model development and score derivation (Table 2).

**Table 2. Multivariable logistic regression analysis.**

| | Full model mOR | 95% CI | P-value | Reduced model mOR | 95% CI | P-value |
|---|---|---|---|---|---|---|
| **Demographic** | | | | | | |
| Male | 0.83 | 0.38–1.79 | 0.630 | | | |
| Age, years | 1.00 | 0.97–1.02 | 0.750 | | | |
| **Symptoms** | | | | | | |
| Myalgia | 1.15 | 0.45–2.94 | 0.773 | | | |
| Jaundice | 0 | 0 | 0.991 | | | |
| Fatigue | 1.48 | 0.70–3.13 | 0.303 | | | |
| Vomiting | 1.20 | 0.51–2.83 | 0.672 | | | |
| Breathing difficulty | 1.53 | 0.55–4.27 | 0.418 | | | |
| **Physical examinations** | | | | | | |
| Body temperature,°C | 0.94 | 0.65–1.35 | 0.722 | | | |
| Pulse rate, per mins | 1.02 | 0.99–1.05 | 0.094 | | | |
| SBP, mmHg | 1.00 | 0.97–1.02 | 0.876 | | | |
| DBP, mmHg | 0.98 | 0.94–1.02 | 0.257 | | | |
| Respiratory rate, per mins | 1.08 | 0.94–1.25 | 0.291 | | | |
| Jaundice | 0 | 0 | 0.990 | | | |
| **Risk factors** | | | | | | |
| Flood or wet ground at home | 1.46 | 0.72–2.95 | 0.290 | | | |
| Flood or wet ground at workplace | 1.82 | 0.67–4.92 | 0.238 | 2.65 | 1.07–6.58 | 0.035 |
| Contact animal water reservoir | 1.82 | 0.89–3.72 | 0.101 | 1.64 | 0.89–3.03 | 0.111 |
| **Laboratory findings** | | | | | | |
| WBC, /μL | 1.00 | 0.99–1.00 | 0.510 | | | |
| Neutrophil, % | 0.95 | 0.87–1.04 | 0.298 | 1.02 | 0.99–1.05 | 0.052 |
| Lymphocyte, % | 0.94 | 0.85–1.05 | 0.283 | | | |
| Monocyte, % | 0.86 | 0.75–0.99 | 0.036 | | | |
| Platelet, /μL | 1.00 | 0.99–1.00 | 0.978 | | | |
| eGFR, mL/min/1.73 m$^2$ | 1.00 | 0.98–1.02 | 0.744 | | | |
| Urine glucose positive | 1.12 | 0.30–4.16 | 0.860 | | | |
| Urine protein positive | 1.23 | 0.59–2.56 | 0.582 | 1.71 | 0.94–3.10 | 0.079 |
| Urine blood positive | 2.02 | 1.01–4.05 | 0.048 | 1.99 | 1.09–3.62 | 0.026 |
| Urine bilirubin positive | 1.77 | 0.44–7.09 | 0.419 | | | |
| Urine ketone | 0.85 | 0.35–2.06 | 0.713 | | | |
| Constant (intercept) | 116.16 | | | 0.02 | | |

Abbreviations: SBP, systolic blood pressure; DBP, diastolic blood pressure; WBC, white blood cell count; eGFR, estimated glomerular filtration rate; mOR, multivariable odds ratio; CI, confidence interval; SD, standard deviation; IQR, interquartile range.

**Table 3. Multivariable logistic model with score transformation via weighing of logit coefficients.**

| Predictors | mOR | 95% CI | P-value | Coefficient | Score |
|---|---|---|---|---|---|
| Wet ground at workplace | 2.66 | 1.06–6.66 | 0.037 | 0.976 | 2 |
| Contact animal water reservoir | 1.73 | 0.94–3.20 | 0.079 | 0.550 | 1 |
| Urine protein positive | 1.99 | 1.09–3.64 | 0.026 | 0.529 | 1 |
| Urine blood positive | 1.70 | 0.93–3.09 | 0.084 | 0.686 | 1.5 |
| Neutrophil count ≥80% | 2.27 | 1.24–4.15 | 0.008 | 0.818 | 1.5 |
| Constant | 0.07 | | | -2.611 | |

Abbreviation: mOR, multivariable odds ratio.

Sequential elimination was carried out based on both statistical significance and model diagnostic performance. Five final predictors remained within the reduced logistic model which were history of exposure to wet ground at workplace, history of contact water reservoir used by animal, urine protein and urine blood positive from dipstick test and neutrophil count from CBC ≥80% (Table 3).

The score was derived by division of larger coefficients with the smallest coefficient resulting in a total score of 7 points. The newly derived diagnostic scoring scheme was named the "OPD Lepto Score". Patients with history of exposure to wet ground at work were assigned 2 points, history of contact with water reservoir used by animal were assigned 1 point, urine protein positive were assigned 1 point, urine blood positive were assigned 1.5 points, and neutrophil count ≥80% were assigned 1.5 points (Table 3). Three pre-specified cut-off points for dichotomization of the score were compared based on sensitivity, specificity, positive likelihood ratio and AuROC (Table 4). The score of 3.5 was chosen based on its AuROC of 0.67 (95%CI 0.61–0.73) and higher sensitivity at 72.4% (95%CI 60.9–82.0) than that of score ≥4. Consequently, suspected patients with OPD score less than 3.5 would be defined as low risk while patient with OPD score ≥3.5 would be defined as high risk of having leptospirosis (Table 5).

The OPD score diagnostic performance was considered acceptable at the AuROC of 0.72 (95% CI 0.65–0.79) (Fig 3). The measure of calibration was done with Hosmer-Lemeshow goodness-of-fit test which yielded non-significant p of 0.637. The diagnostic ability of OPD score with 5 predictors was then compared with that of WHO score with 10 predictors, the test of equality revealed statistically significant differences of AuROC between the two scores (0.72 vs 0.62, p = 0.014). The bootstrapped ROC from internal validation of OPD score was 0.67 (95%CI 0.59–0.74). In a clinical setting where complete blood count was unavailable, the OPD score still could maintain its diagnostic performance at the AuROC of 0.70 for the remaining 4 predictor variables.

## Discussion

In early stage of leptospirosis infection, the clinical signs and symptoms were found to be indifferent and did not provide significant diagnostic value. In this study, we explored for potential predictors for diagnosis of early leptospirosis infection in terms of initial clinical characteristics,

**Table 4. Selection of score cut point, sensitivity, specificity, LHR+, AuROC.**

| Score cut point | Sensitivity (%) | Specificity (%) | LHR+ | AuROC(95%CI) |
|---|---|---|---|---|
| <3 | 86.8 (77.1–93.5) | 39.5 (32.1–47.4) | 1.44(1.24–1.67) | 0.63 (0.58–0.69) |
| ≥3.5 | 72.4 (60.9–82.0) | 61.7 (54.8–69.1) | 1.89 (1.49–2.39) | 0.67 (0.61–0.73) |
| ≥4 | 64.5 (52.7–75.1) | 71.3 (63.8–78.0) | 2.24 (1.68–3.00) | 0.68 (0.61–0.74) |

AuROC, area under the receiving operating curve

**Table 5. Score categorization and likelihood ratio of positive (LHR+) in OPD Lepto Score and Faine's score.**

| Probability categories | Score | Confirmed cases (n = 75) | | Non-cases (n = 167) | | LHR+ | (95%CI) | P-value |
|---|---|---|---|---|---|---|---|---|
| | | n | % | n | % | | | |
| OPD Lepto score | | | | | | | | |
| Low | <3.5 | 21 | 28.0 | 103 | 61.7 | 0.45 | (0.31–0.67) | <0.001 |
| High | ≥3.5 | 54 | 72.0 | 64 | 38.3 | 1.88 | (1.48–2.39) | <0.001 |
| Mean±SD | - | 4.2 | ±1.5 | 3.0 | ±1.6 | - | | <0.001 |
| Faine's' score | | | | | | | | |
| Low | <26* | 97.3 | 76.0 | 165 | 98.8 | 0.99 | (0.95–1.03) | 0.407 |
| High | ≥26* | 2 | 2.7 | 2 | 1.2 | 2.23 | (0.32–15.5) | 0.407 |
| Mean±SD | - | 19.55 | ±3.2 | 17.7 | ±4.01 | - | | 0.002 |

*Cut point of the conventional Faine's' score is definition of a presumptive diagnosis of leptospirosis from Part A, or Parts A and B score [21]

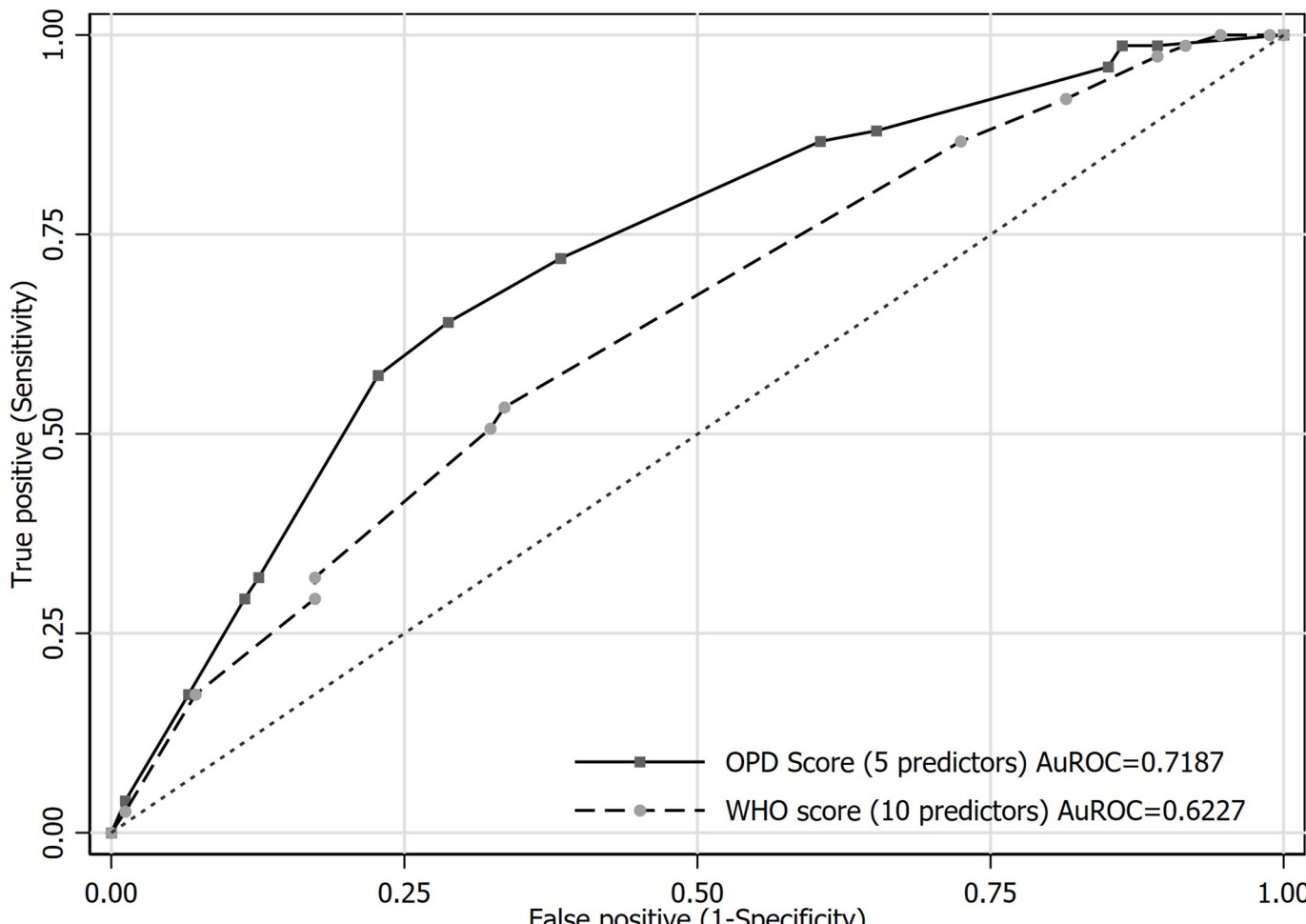

**Fig 3. Receiver operating characteristic curve in diagnostic prediction of leptospirosis, comparison of OPD score and standard WHO score within development cohort.**

history of environmental exposure to the organism and basic laboratory investigations which could be done routinely. We identified five final predictor variables which includes mainly the environmental exposure risk and basic routinely-available laboratory investigation. A history of wet ground at workplace, contact with animal water reservoir, positive urine protein and urine blood from dipstick test, and a neutrophil count of more than 80% from complete blood count were included in the multivariable model for diagnosis of early leptospirosis patients.

The original Faine's criteria which was widely endorsed since 1982 consisted of three parts which were clinical data, epidemiological factors and bacteriological and laboratory findings [21]. Nine predictor variables were needed from patient's clinical presentation such as headache, fever, body temperature 39 ≥degree Celsius, conjunctival suffusion and meningism. For epidemiological factors, the patients were asked for history of contact with animals or contaminated water. Two types of laboratory investigation, culture and MAT, were necessary to fulfil the criteria. In 2004, the criteria were revised to improve applicability in many endemic areas [22–24]. The Modified Faine's 2004 added rainfall as another important epidemiological factors and split animal contact and contact with contaminated environment into two independent factors. ELISA test for IgM and slide agglutination test or SAT were appended as serology options. Eight years later, the latest version of the criteria was launched as Modified Faine's criteria 2012 [25]. The criteria added hemoptysis and dyspnea in clinical data and polymerase chain reaction as another option for laboratory investigation. All the criterion were based on acutely-ill patients with typical clinical syndrome and they also required advanced laboratory testing which was usually unavailable especially in health resource limitted areas.

After a widespread leptospirosis endemic in Thailand during 1997–1998, the Faine's criteria was implemented for use in high risk areas specifically in the northeastern and southern region of the country. The bacteriological and laboratory findings component of the score was not completed owing to financial and logistic limitations. Only few specimens were sent to the central laboratory investigation for epidemiologic purposes. The diagnostic performance varied and was unsatisfied with 68% sensitivity and 58% specificity [31]. For hospitalized patients, who usually presented with severe clinical symptoms, an on-admission diagnostic tool (THAI-LEPTO Score) was developed based on in-patient data from local hospitals within the northeastern and southern part of Thailand [32]. The score contains 8 clinical parameters which would be present in leptospirosis patients with end-organ damage. Although the predictive performance was good, the test might not be sensitive for use in out-patient settings where early phase leptospirosis patients usually present with mild and vague clinical syndromes.

In this study, clinical presentations, both signs and symptoms, did not reveal any significant differences and thus might not be efficient to consider as potential predictors. As patients with acute febrile illness in the tropical regions usually presented in the early phase with fever without any distinctive features to distinguish leptospirosis from other types of infections [9, 33–34]. Since leptospirosis infection required environmental exposure to *leptospira*, history of exposure to potential risk factors were crucial. It was concordantly found that history of flooded or wet ground at workplace was a good predictor [35–37]. Patients with flooded workplace had high probability of exposure to infective organism for longer duration than other places. Patients with flooded house compound also carried some risk but not as high as at workplace because people tend to reside within dry area at home. Contact or utilize animal water reservoir was another predictor within the OPD Lepto Score. In agricultural area within the northeastern, animal water reservoirs were usually natural sources without steep curb. Animals and livestock can drink and bath within this basin. Patients who contacted or utilized the water within same natural water reservoirs had higher risk of exposure to *leptospira*.

Presence of protein and blood in the urine of suspected patients was due to renal involvement of leptospirosis, which could be one of the most distinguish features from other tropical

**Table 6. Comparison of study characteristics and diagnostic indices between previous diagnostic criteria of leptospirosis and newly derived OPD Lepto Score.**

| | Chifou W. et al.,1997[31] | Bhatia M. et al.,2015[23] | Bandara K et al., 2016[25] | Linda Rose Jose. et al., 2016[24] | Temeiam N. et al. |
|---|---|---|---|---|---|
| Criteria or score | Faines' criteria 1982 | Modified Faine's criteria (2004) | Modified Faine's criteria (with amendment) 2012 | Modified Faine's criteria (2004) | OPD Lepto Score |
| Country | Thailand | India | Sri Lanka | India | Thailand |
| Reference standard | MAT titer ≥ 400 or 4 fold raising of MAT paired serum | MAT titer ≥ 1:80 | MAT1≥400 or PCR | IgM ELISA (PanBio, Brisbane, Australia) and MAT1>1:100 | Culture or PCR or MAT1 ≥ 800 or 4-fold raising MAT2 |
| Patient domain | Hospitalized 54.1% and non-hospitalized 45.9% | Hospitalized | Hospitalized | Hospitalized | Outpatient clinic |
| Sample size (suspected/ confirmed) | 74 /24 | 100/49 | 168/66 | 332 /115 | 260/82 |
| Sensitivity (%) | 68 | N/A | 89.39 | N/A | 72.4 |
| Specificity (%) | 58 | N/A | 58.82 | N/A | 61.7 |
| PPV (%) | 64 | 21 | 58.42 | N/A | 46.2 |
| NPV (%) | 59 | N/A | 89.55 | N/A | 83.1 |
| Recommendations | Urine protein≥1+ plus fever, headache, and myalgia increase specificity to 80% * cutoff point ≥ 20 | Further evaluation of the diagnostic utility of modified Faine's criteria is needed of the hour. | Utilized only immunochromatographic assay (Leptocheck WB, India) in Part C be useful in the presumptive diagnosis of leptospirosis. | All persons with fever for >5 days should be screened for leptospirosis utilizing modified Faine's criteria | Two risk factors, protein and blood urine dipstick, and neutrophil count ≥ 80% |

Abbreviations: MAT, microscopic agglutination test; PCR, polymerase chain reaction; N/A, not applicable.

diseases. *Leptospira* directly invades the vessels causing vascular damages and stimulate a chain of inflammatory processes. Kidneys are the most commonly involved organs [1, 38], thus, even in the early stage of the disease, urine abnormalities from dip stick test can be positive. The last predictor was percentage of neutrophil count ≥80% [39–40]. Infectious disease with higher percentage of neutrophil could indicate bacterial origin, while other tropical diseases were caused by either viral or atypical organism such as rickettsia.

Five potential predictors constituted a total score of seven with acceptable diagnostic performance according to Hosmer-Lemeshow categorization of AuROC. Comparing to standard WHO score or conventional Faine's Criteria, the OPD Score had higher diagnostic ability with fewer number of predictors (Table 6).

In the OPD Lepto Score implementation, we chose the cut-off at ≥ 3.5 points because the sensitivity and specificity were in acceptable margin and not too low to cause error in diagnosis. Other factors should be considered together with the use of the score in real practice. The score was designed to be used in undifferentiated fever patients intended to be diagnosed of leptospirosis, so the patients' signs and symptoms must be relevant. Timing and season of visit should also be taken into account prior to antibiotic prescription. Patients who scored low risk during low prevalence season should be closely monitored by village health volunteers or local health care personnel for disease progression, while patients who scored high risk could be offer a choice of antibiotics such as doxycycline. Although a portion of high-risk patients might receive unnecessary antibiotics, the benefit certainly outweighed the risk for this group of patients [41]. After treatment initiation, it is mandatory to schedule the patient for evaluation.

Another obstacle that delay patients visit to the hospital occurred during harvesting season, where outbreaks usually took place. Large portions of high-risk patients, the farmers, avoided distant traveling to hospital by visiting primary care units within their villages. The primary

care units did not have attending physician and were not equipped with laboratory investigation other than urine drip stick test. Local health care workers could exploit the remaining 4 predictors of OPD score as a guide in initiation of early antibiotic treatment by leaving out neutrophil percent count and still preserved a diagnostic performance of 0.70. This could help alleviate disease progression and probably decrease fatality rate of leptospirosis.

Our study has several strengths. First, this is a prospective study with data collected from multiple sites within the leptospirosis endemic regions in the northeastern parts of Thailand. The study was primarily designed for patients who presented with undifferentiated fever and physicians suspected of having leptospirosis. Second, the domain of patients was shifted from clear clinical syndromes as in other derived score (e.g. Faine's Criteria or WHO Score, and THAI-LEPTO Score) to patients with vague symptoms for this purpose. We believed that early diagnosis and early treatment in suspected group of patients could impede disease progression and decrease mortality. Third, the included predictors used in OPD Lepto Score were based on routinely available laboratory investigations and exposure to risk factors could be simply asked within short time.

However, there were some limitations need to be addressed. First, the study protocol planned to schedule all enrolled patients for 2nd visit for blood samples for MAT to observe and interpret four-fold rising of convalescent titers, but only 48.9% of patients came for the blood tests. Thus, the number of confirmed cases could be underestimated in some degree. Second, distinguishing a group of patients at their early phase of the disease was troublesome and resulting in only fair discriminative performance of the model. However, we believed that as the model was derived entirely from the specific study domain intended to be used in outpatient practice, its performance should overcome previously developed criterion. Third, in terms of generalizability, the development cohort was based on highly endemic region of leptospirosis, so the application of OPD Lepto Score to non-endemic area might not be beneficial. Fourth, an external validation study is needed for evaluation of generalizability before the OPD Lepto Score being endorsed for real clinical use. Fifth, the confirmation of diagnosis other than leptospirosis in non-cases was not done. These some non-cases may be actual cases but remained undiagnosed because of various reasons including lesser severity which lowers diagnostic techniques sensitivity, higher antibiotic prescription (even non-significantly). These might increase the risk of false negatives.

In conclusion, this study developed and internally validated the OPD Lepto Score, a practical clinical risk score for the diagnosis of leptospirosis in suspected patients with acute undifferentiated fever who presented to the outpatient clinics in high endemic areas. With 5 predictors, the score was more practical for outpatient setting than the conventional WHO score with 10 predictors.

## Supporting information

**S1 Table. Univariable analysis of baseline demographic data, presenting symptoms, physical examinations and initial laboratory investigations of confirmed-cases and non-cases of Leptospirosis.**
(DOCX)

## Acknowledgments

The authors wish to thank the following for their contributions.

■ Directors and officers of Khukhan, Khun Han, Phu Sing, Phrai Bueng, and Prang Ku Hospitals and the District Public Health Offices for data collection.

- Local health care workers in all health promotion hospitals appointing patients for second blood sampling visit.

- Si Sa Ket Provincial Public Health Office for their co-ordinations.

- Department of disease prevention and epidemiology, Si Sa Ket Hospital for data collection and coordination with central laboratory for serological testing.

- Center for Critical Care Nephrology, The CRISMA Center, Department of Critical Care Medicine, University of Pittsburgh School of Medicine, Pittsburgh, Pennsylvania, United States of America for specimen culture and polymerase chain reaction testing.

- Department of Medical Sciences, Ministry of Public Health, THAILAND for microscopic agglutination testing (MAT).

## Author Contributions

**Conceptualization:** Nidhikul Temeiam, Sutthi Jareinpituk, Nattachai Srisawat.

**Data curation:** Nidhikul Temeiam, Sutthi Jareinpituk.

**Formal analysis:** Nidhikul Temeiam, Sutthi Jareinpituk, Phichayut Phinyo, Jayanton Patumanond.

**Investigation:** Nidhikul Temeiam, Sutthi Jareinpituk, Phichayut Phinyo, Jayanton Patumanond, Nattachai Srisawat.

**Methodology:** Nidhikul Temeiam, Sutthi Jareinpituk, Phichayut Phinyo, Jayanton Patumanond, Nattachai Srisawat.

**Project administration:** Nidhikul Temeiam.

**Resources:** Nattachai Srisawat.

**Software:** Phichayut Phinyo, Jayanton Patumanond.

**Supervision:** Nattachai Srisawat.

**Validation:** Phichayut Phinyo, Jayanton Patumanond.

**Visualization:** Phichayut Phinyo, Jayanton Patumanond.

**Writing – original draft:** Nidhikul Temeiam, Sutthi Jareinpituk.

**Writing – review & editing:** Phichayut Phinyo, Jayanton Patumanond, Nattachai Srisawat.

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
