## [Decision Letter · Decision Letter 0]

14 Oct 2019

Dear Dr. Srisawat:

Thank you very much for submitting your manuscript "DEVELOPMENT and VALIDATION of a SIMPLE SCORE for DIAGNOSIS of LEPTOSPIROSIS at OUTPATIENT DEPARTMENTS" (#PNTD-D-19-01397) for review by PLOS Neglected Tropical Diseases. Your manuscript was fully evaluated at the editorial level and by independent peer reviewers. The reviewers appreciated the attention to an important problem, but raised some substantial concerns about the manuscript as it currently stands. These issues must be addressed before we would be willing to consider a revised version of your study. We cannot, of course, promise publication at that time.

We therefore ask you to modify the manuscript according to the review recommendations before we can consider your manuscript for acceptance. Your revisions should address the specific points made by each reviewer. 

When you are ready to resubmit, please be prepared to upload the following:

(1) A letter containing a detailed list of your responses to the review comments and a description of the changes you have made in the manuscript.

(2) Two versions of the manuscript: one with either highlights or tracked changes denoting where the text has been changed (uploaded as a "Revised Article with Changes Highlighted" file); the other a clean version (uploaded as the article file).

(3) If available, a striking still image (a new image if one is available or an existing one from within your manuscript). If your manuscript is accepted for publication, this image may be featured on our website. Images should ideally be high resolution, eye-catching, single panel images; where one is available, please use 'add file' at the time of resubmission and select 'striking image' as the file type. 

Please provide a short caption, including credits, uploaded as a separate "Other" file. If your image is from someone other than yourself, please ensure that the artist has read and agreed to the terms and conditions of the Creative Commons Attribution License at http://journals.plos.org/plosntds/s/content-license (NOTE: we cannot publish copyrighted images). 

(4) If applicable, we encourage you to add a list of accession numbers/ID numbers for genes and proteins mentioned in the text (these should be listed as a paragraph at the end of the manuscript). You can supply accession numbers for any database, so long as the database is publicly accessible and stable. Examples include LocusLink and SwissProt.

(5) To enhance the reproducibility of your results, we recommend that you deposit your laboratory protocols in protocols.io, where a protocol can be assigned its own identifier (DOI) such that it can be cited independently in the future. For instructions see http://journals.plos.org/plosntds/s/submission-guidelines#loc-methods

While revising your submission, please upload your figure files to the Preflight Analysis and Conversion Engine (PACE) digital diagnostic tool, https://pacev2.apexcovantage.com/ PACE helps ensure that figures meet PLOS requirements. To use PACE, you must first register as a user. Then, login and navigate to the UPLOAD tab, where you will find detailed instructions on how to use the tool. If you encounter any issues or have any questions when using PACE, please email us at figures@plos.org.

We hope to receive your revised manuscript by Dec 13 2019 11:59PM. If you anticipate any delay in its return, we ask that you let us know the expected resubmission date by replying to this email.

To submit a revision, go to https://www.editorialmanager.com/pntd/ and log in as an Author. You will see a menu item call Submission Needing Revision. You will find your submission record there. 

Sincerely,

Nicholas P. Day

Associate Editor

Mathieu Picardeau

Deputy Editor

Reviewer's Responses to Questions

**Key Review Criteria Required for Acceptance?**

**Methods**

-Are the objectives of the study clearly articulated with a clear testable hypothesis stated?

-Is the study design appropriate to address the stated objectives?

-Is the population clearly described and appropriate for the hypothesis being tested?

-Is the sample size sufficient to ensure adequate power to address the hypothesis being tested?

-Were correct statistical analysis used to support conclusions?

-Are there concerns about ethical or regulatory requirements being met?

Reviewer #1: (No Response)

Reviewer #2: (No Response)

Reviewer #3: Authors have developed a score used for as a diagnosis tool to identify mild or early phase symptoms of human leptospirosis in rural healthcare units with limited laboratory resources based on data from "cases" and "non-cases" patients in Si Sa Ket, a province in Northeast Thailand, between December 2017 and November 2018. 

Most methods were lacking of reference citations for example, all laboratory tests used in the study such as culture, polymerase chain reaction, and MAT methods; cases and non-cases definitions; the method used for sample size estimation; and the modified WHO score used in the study.

Section "Blood sample and specimen collection" is not well described. For example, which kinds of blood tubes were used for culture, PCR and MAT, which parts of blood (or urine) samples were collected after centrifuge and for what tests, centrifugation force was not specified. It is confusing that frozen samples were sent to culture (this definitely need the step-by-step-protocol or reference citation.

For the definition of leptospirosis cases, the first criteria "isolation of Leptospira from clinical specimen" might not accurate. How much do authors be certain if pathogenic Leptospira species isolated from urine samples collected from patients who having 2 days of fever? Or how much author be certain the species or group (pathogens or non-pathogens) of Leptospira isolated from blood or urine specimens. 

Clinical outcomes (at least after 7 days, the second visit) and treatments are important for evaluation the success of non-intervention routine practices. However, these data were not collected in this study.

**Results**

-Does the analysis presented match the analysis plan?

-Are the results clearly and completely presented?

-Are the figures (Tables, Images) of sufficient quality for clarity?

Reviewer #1: (No Response)

Reviewer #2: (No Response)

Reviewer #3: The results of Leptospira isolations (both blood and urine specimens), PCR and MAT (paired serum) were not presented. These results are keys results for proper "cases" and "non-cases" definitions. Only final number of cases and non-cases were described.

The results of WHO score in each "cases" and "non-cases" groups are important, however, they were not shown. 

Table 1 and 3 shows combination of both category parameters and numeric parameters. However, the heading are not specify well. 

Is it accurate to use mean (in stead of median and IQR) to compare the OPD score of "cases" and "non-cases" group as shown in Table 6?

**Conclusions**

-Are the conclusions supported by the data presented?

-Are the limitations of analysis clearly described?

-Do the authors discuss how these data can be helpful to advance our understanding of the topic under study?

-Is public health relevance addressed?

Reviewer #1: (No Response)

Reviewer #2: (No Response)

Reviewer #3: The data shown in the manuscript supported that "the OPD score" is better than the WHO score in the study endemic setting of Si Sa Ket province. However, the discussion on why the WHO score is not successful when used in this setting is not mention (even Laboratory data was available). Is the OPD score is suitable for human leptospirosis with other complication such as respiratory or heart diseases (not only kidney complication). Or is the OPD score is suitable for other endemic area in the Southern of Thailand or Malaysia, where less cattle farms and rice paddy fields but more rubber gardens, forest trekking and stray dogs.

**Editorial and Data Presentation Modifications?**

Reviewer #1: (No Response)

Reviewer #2: (No Response)

Reviewer #3: 1. Revision the "Methods" with all possible reference citations and laboratory brief protocols. And also specify the following details; PCR format (conventional or real-time PCR), target gene (rrs, lipL32 or others), Laboratory institute where all tests performed. 

2. Adding PCR, culture and MAT results; and the WHO score results. 

3. Improving heading and legend in Table 1 and 3.

4. Using median and IQR instead of mean and SD when it is appropriate.

5. Improving discussion on the limitation of the "OPD score".

6. Checking spelling and grammar for the whole manuscript again. Please correct the "drip stick". It should be "dipstick".

**Summary and General Comments**

Reviewer #1: (No Response)

Reviewer #2: Manuscript Review: Development and Validation of a Simple Score for Diagnosis of Leptospirosis at Outpatient Departments 

Journal: PLoS NTD

Manuscript ID: PNTD-D-19-01397

This study proposes a scoring system for identifying patients with early leptospirosis. The rationale for the study is that identification of early leptospirosis using a simple set of questions and tests would be broadly applicable and could enable therapy at a stage that could prevent or reduce more severe complications. Although cases of leptospirosis were confirmed by MAT, serologic testing and other leptospirosis-specific tests were not considered as these are not available in many settings. 

An OPD score was developed based on five relatively novel and potentially useful criteria. The “wet ground at workplace” criterion is certainly more specific than the Modified Faine’s criterion of “rainfall”. Likewise, contact with an “animal water reservoir” is more specific than “animal contact” or contact with “contaminated environment”. The three laboratory criteria are readily obtainable in many outpatient settings. Although the proposed OPD scoring system could be potentially useful, I have a number of concerns that should be addressed.

Comments:

1. The authors should summarize Faine’s original and modified criteria for readers not familiar with these diagnostic scoring systems.

2. The authors should summarize the sensitivity and specificity of Modified Faine’s Criteria as reported by others (references 24-27) and compare these published results with those reported here.

3. The crude odds ratios for “wet ground at workplace” of 2.83 is not supported by the percentages in confirmed (27.4%) and non-cases (20.2%).

4. How much protein or blood in the urine is required for a positive score for those criteria?

5. The data for patients with neutrophil percentage >80% among confirmed and non-cases is missing.

6. The data for some of the modified Faine’s criteria (such as rainfall) among confirmed and non-cases is missing.

7. What percentage of non-cases were diagnosed with infections other than leptospirosis? What were those alternative diagnoses and how often did they occur?

8. Did any of the leptospirosis patients in this study develop more severe complications including death? How many of the leptospirosis cases in this study were later admitted to the hospital or treated for leptospirosis?

9. What was the average time from onset of symptoms to evaluation?

Reviewer #3: The OPD score is an alternative tool for screening human leptospirosis for rural healthcare units with limited laboratory resources. However, further sensitivity and specificity validations in other endemic (various prevalence) and non-endemic regions should be emphasized.

PLOS authors have the option to publish the peer review history of their article (what does this mean?). If published, this will include your full peer review and any attached files.

Reviewer #1: No

Reviewer #2: No

Reviewer #3: No

---

## [Editor Report · Decision Letter 1]

8 Dec 2019

Dear Dr. Srisawat,

We are pleased to inform you that your manuscript, "DEVELOPMENT and VALIDATION of a SIMPLE SCORE for DIAGNOSIS of LEPTOSPIROSIS at OUTPATIENT DEPARTMENTS", has been editorially accepted for publication at PLOS Neglected Tropical Diseases.

Before your manuscript can be formally accepted and sent to production you will need to complete our formatting changes, which you will receive in a follow up email. Please note: your manuscript will not be scheduled for publication until you have made the required changes.

IMPORTANT NOTES

* Copyediting and Author Proofs: To ensure prompt publication, your manuscript will NOT be subject to detailed copyediting and you will NOT receive a typeset proof for review. The corresponding author will have one final opportunity to correct any errors when sent the requests mentioned above. Please review this version of your manuscript for any errors.

* If you or your institution will be preparing press materials for this manuscript, please inform our press team in advance at plosntds@plos.org. If you need to know your paper's publication date for media purposes, you must coordinate with our press team, and your manuscript will remain under a strict press embargo until the publication date and time. PLOS NTDs may choose to issue a press release for your article. If there is anything that the journal should know, please get in touch.

*Now that your manuscript has been provisionally accepted, please log into EM and update your profile. Go to http://www.editorialmanager.com/pntd, log in, and click on the "Update My Information" link at the top of the page. Please update your user information to ensure an efficient production and billing process.

*Note to LaTeX users only - Our staff will ask you to upload a TEX file in addition to the PDF before the paper can be sent to typesetting, so please carefully review our Latex Guidelines [http://www.plosntds.org/static/latexGuidelines.action] in the meantime.

Best regards,

Nicholas P. Day

Associate Editor

Mathieu Picardeau

Deputy Editor

---

## [Editor Report · Acceptance letter]

2 Jan 2020

Dear Dr. Srisawat,

We are delighted to inform you that your manuscript, "DEVELOPMENT and VALIDATION of a SIMPLE SCORE for DIAGNOSIS of LEPTOSPIROSIS at OUTPATIENT DEPARTMENTS," has been formally accepted for publication in PLOS Neglected Tropical Diseases.

Best regards,

Serap Aksoy

Editor-in-Chief

Shaden Kamhawi

Editor-in-Chief
